



# Approaches to calibrate *in-situ* capacitance soil moisture sensors and some of their implications

N. A. L. Archer [1], B. R. Rawlins[2], B. P. Marchant[2], J. D. Mackay[2] and P. I. Meldrum[2]

[1]British Geological Survey, Lyell Centre, Research Avenue South, Edinburgh EH14 4AP

[2]British Geological Survey, Environmental Science Centre, Nicker Hill, Keyworth, Nottingham, NG12 5GG

*Correspondence to*: Nicole A. L. Archer (nicarc@bgs.c.uk)

**Abstract.** Capacitance probes are increasingly being used to monitor volumetric water content (VWC) in field conditions and are provided with in-built factory calibrations so they can be deployed at a field site without the requirement for local calibration. These calibrations may not always have acceptable accuracy and therefore to improve the accuracy of such calibrations soil-specific laboratory or field calibrations are required. In some cases, manufacturers suggest calibration is undertaken on soil in which the structure has been removed (through sieving or grinding), whilst in other cases manufacturers suggest structure may be retained. The objectives of this investigation were to i) demonstrate the differences in laboratory calibration of the sensors using both structured and unstructured soils, ii) compare moisture contents at a range of suctions with those predicted from soil moisture release curves for their texture classes iii) compare the magnitude of errors for field measurements of soil moisture based on the original factory calibrations and the laboratory-based calibrations using structured soil.

 Grinding and sieving clay soils to <2mm and then repacking the clay to bulk densities similar to *in-situ* field bulk densities was found not to represent the same field conditions for accurate VWC. When adding >50% water to the ground and sieved soil samples, dielectric values to VWC >50% were observed to be significantly lower than using undisturbed soil cores taken from the field and therefore undisturbed soil cores were considered to be better to calibrate capacitance probes. Generic factory calibrations for most soil sensors have a range of measurement from 0 to 50%, which is not appropriate for the studied clay-rich soil, where ponding can occur during persistent rain events, which are common in temperate regions.

**Keywords**. Volumetric water content, capacitance sensors, clay-rich soils, soil water release curves, soil shrinkage

## 1 Introduction

Field-based*, in situ,* automatic sensors and networks formed from them, are increasingly used to monitor soil moisture, and provide unique possibility to investigate spatial and temporal soil water dynamics (Vereecken et al., 2014), which is important for a range of applications related to the hydrological cycle including agricultural production, meteorology and groundwater recharge.

One of the most popular soil moisture sensors is the capacitance probe and is based on functional relationships established between soil moisture content and dielectric permittivity ($\varepsilon$), which can be determined for a small volume of soil using a series of probes (e.g. ML3 ThetaProbe, Delta-T devices Ltd, Cambridge, UK, 5TE sensor Decagon Devices Inc, Pullman, US and CS658, Campbell Scientific Inc., Logan, US). Such sensors are provided

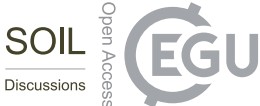

with in-built factory calibrations so they can be deployed at a field site providing data on volumetric moisture
content, without the requirement for local calibration. The commercial companies who supply the sensors provide
uncertainties for these calibrations. For example, Decagon Devices (2014) suggest accuracies ± 3% of volumetric
water content (VWC) for the 5TE sensor using a generic calibration in mineral soils that have a solution electrical
conductivity <10dS/m, Delta-T Devices(2016) specify that the ML3 ThetaProbe has ± 1% accuracy using three
different generic soil calibrations depending on soil type and soil salinity <0.034 $m^3$ $m^{-3}$ and Campbell Scientific
(2016) state ± 2.5% accuracy for the CS616/625s standard factory calibration when bulk soil EC is ≤0.5 $dSm^{-1}$ and
soil bulk density is ≤1.55 g $cm^{-3}$. All three soil moisture sensors have a VWC measurement range between 0% and

9   50%.

Of the few published studies that have assessed the accuracy of the factory calibrations, Varble and Chávez (2011),
conclude that the factory calibrations of capacitance probes, such as the 5TE Decagon Devices, Inc., Pullman, WA
and TDT, Acclima, Inc., Meridian, ID are acceptable for the sandy clay loam in applications that do not need high
accuracy, however for soils that have more clay/silt content, a field-based calibration is recommended. Luis
Gabriel et al. (2010) on the other hand recommends soil-specific calibrations are needed only if users are interested
in absolute values, but  not necessary for relative differences.
To improve capacitance probe accuracy for soil water measurement, soil-specific laboratory or field calibrations
are required (Blonquist et al., 2005; Bogena et al., 2007; Evett et al., 2006; Luis Gabriel et al., 2010; Parvin and
Degre, 2016; Varble and Chavez, 2011). To calibrate capacitance sensors it is necessary to understand the
relationship between the dielectric constant (Ɛ) and soil moisture. This is done by using standard dielectric liquids
e.g. (Bogena et al., 2007; Jones et al., 2005; Rosenbaum et al., 2010) or soil-specific calibrations, e.g. (Chanzy et
al., 1998; Luis Gabriel et al., 2010; Varble and Chavez, 2011). Soil-specific calibrations can be achieved in the
laboratory or field site by modifying moisture content and using gravimetric approaches to determine a range of
soil moisture and associated Ɛ values. In some cases, manufacturers suggest calibration is undertaken on soil in
which the structure has been removed (through sieving or grinding) (Campbell Scientific Inc., 2016; Cobos and
Chambers, 2010; Parvin and Degre, 2016), whilst in other cases manufacturers suggest structure may be retained
(Delta-T Devices Ltd., 2013).  There are three reasons we considered that sensor calibrations should ideally be
undertaken using structured soils.  First, removal of structure could lead to unreliable results when sensors are
deployed in structured soils in the field. Second,  the importance of soil structure in determining pore size
distributions (and associated matric potentials; (Nimmo, 2004)). Third,  the so called 'sphere of influence' (the
soil volume around the probe's electrodes influencing the probe's measurement), which is typically small for
capacitance probes (Chanzy et al., 1998), making them sensitive to small scale variations in soil structure and soil
water content near to the probe electrodes (Evett et al., 2006).
To our knowledge, there has been little or no research to date on the magnitude of any error and (or) systematic
bias introduced into soil moisture sensor measurements using either: i) factory calibrations; or ii) calibrations based
on soils from which the structure has been removed. This has important implications because practitioners may be
unaware of these potential errors.
Such practitioners may also rely on pedotransfer functions (PTFs) derived from relatively large databases for soils
of differing texture class and bulk density (e.g. ROSETTA (Schaap et al., 2001)) to provide a certain amount of



validation for the initial measurements from their moisture sensors. However, PTFs may also be prone to error
and (or) bias and could provide false confirmation of erroneous sensor outputs. For example, Hodnett and
Tomasella (2002) found that PTFs developed from a temperate dataset had a poorly represented clay textural class,
which resulted in errors for predicting water retention curves for clay soils. To provide an absolute understanding
of soil moisture relationships across a range of hydrological conditions it is often necessary to measure soil
moisture release characteristic curves using a pressure plate (Klute, 1986).
In this paper we present our findings from laboratory measurements to determine the accuracy of factory
calibrations for a capacitance probe in clay dominated soils at a field site in northern England. We demonstrate
the differences in laboratory calibration of the sensors using both structured and unstructured soils. We compare
moisture contents at a range of suctions with those predicted from soil moisture release curves for their texture
classes. Finally, we compare the magnitude of errors for field measurements of soil moisture based on the original
factory calibrations and the laboratory-based calibrations using structured soil. We summarize our findings and
discuss their implications for the use of Ɛ-based soil moisture sensors.

## 2    Site description

The study area is located 12 km west of Malton, North Yorkshire, UK, and is part of the south facing Hollin Hill
escarpment. It provides rough grazing through-out the year for approximately 30 sheep and is a landslide research
site monitored by the Automated Time Lapse Electrical Resistivity Tomography (ALERT) system, as described
by Wilkinson, et al. (2010) and Chambers et al. (2011). The area is considered as a representative landslide site,
typically associated to the Lias Group mudrocks (Hobbs et al., 2005) and is described in detail by Gunn, et al.

20    (2013).

The use of 3D Electrical Resistivity Tomography methods has detailed the contrasting layers of the weathered
mudrock sliding above the more resistive, permeable layers of coarser grained silt and sandstones, causing zones
of depletion and accumulation of superficial clay, silts and sandy materials, as described by Chambers et al (2011).
The combination of the Lias Group formations and landslide processes creates a slope of contrasting soil textures
where the upper part of the slope is dominated by clays relating to the Whitby Mudstone formation and the lower
slope is dominated by sand derived from the Straithes Sandstone and Cleveland Ironstone Formations (Hobbs et
al., 2005). The zones of rotational landslide failure and zones of accumulation have developed active uneven
landslide lobes across the slope. The most recent failures are shown as features (fig. 1) of an along-contour
backscar and then below zones of hummocky ground downslope, while broken-up and annealed materials from
former older lobe advances are present at the bottom of the slope, as low-lying lobate humps, which extend beyond
the lower part of the field. (Gunn et al., 2013).

## 3    Methodology

Dominant soil characteristics of the field area were first determined to understand the dominant soil types and
where capacitance probes should be located within the field site. As the site was dominated with clay soils, soil
shrinkage was measured to determine the significance of soil shrinkage of soil cores taken from the field and used





to calibrate the capacitance probes. Water release curves were also created using the pressure chamber technique
to obtain saturation, field capacity and perminant wilting points for each soil type. This also created reference
values to verify pedotransfer functions and calibration curves. Finally, calibration equations for each dominant soil
type were developed using two laboratory techniques: 1) Disturbed calibration method, which ground and sieved
soil taken from the field site and 2) Undisturbed claibration method, which took relatively large undisturbed soil
cores from the field area, maintaining soil structure. The resulting calibration equations for each dominant soil type
and calibration method, were then compared to the measured water release curves and VWC observed in the field
at the same time as field capacitance probe measurements.

### 3.1    Soil characteristics and sensor layout

Eight replicate soil samples at 0.10 m were taken from the numbered positions in fig. 1. Dry mass, bulk density,
soil texture and organic matter (loss on ignition) were measured. From areas 1 to 4 (shown on fig. 1), soil texture
at 10cm were all 100 % clay, except one sample in area 1, which had 71 % clay, 20 % silt and 8 % sand. Organic
matter by loss–on-ignition (maximum temperature of furnace 375 ℃) ranged from 6% to 11% in areas 1 to 4. At
the bottom of the slope, the clay content decreased and sand content increased particularly in areas 5 and 6 (fig.
1), whereas in areas 7 and 8, clay and silt content increased and sand decreased. This data was combined and
plotted in fig. 2, showing the particle size distributions and their  texture classes for the study areas A (clay), B
(sandy clay loam) and C (sandy clay), which are located in fig. 1.
To find the boundary line where the shallow soil changes (0 to 0.4 m depth), a vegetation survey was undertaken
to investigate variations in grass associations that may relate to changes in soil. There were three main changes of
grass associations across the site slope: 1) the upper slope was mainly dominated by *Poa* and *Lolium* genii; 2)
below the landside lobes on the western side of the slope and along the south east edge of the site, was dominated
by *Poa*, *Festuca* and *Holcus lanatus*; and 3) *Holcus lanatus* became sparse on the lower slope below the most
eastern landslide lobe and *Poa* and *Festuca* continued to dominate. Following these vegetation associations, an
auger survey to 0.4 meters depth was undertaken along the boundaries of these changing grass genii associations
to determine the main areas where soil texture changed. The resulting soil texture map was developed (fig. 1),
illustrating that the boundary change of soil texture relates to the landslide lobes. In the lower east region more
visible fresh slumps are observed than the west side of the slope, suggesting that the east side of the slope has more
recent slumping events compared to the west side of the slope (Merritt et al., 2014). The presence of soils with
sandy clay classification on the east side of the slope occurs in the more recently slumped areas, as shown in fig.
1. The soils with larger proportions of sand-sized fraction  occur on the older flow deposits, which is attributed to
leaching and removal of clays infiltrating with rainfall (Merritt et al., 2014).
Following the site survey three areas were chosen to take into account differences in soil properties (fig. 1). These
areas were Group A located on clay, Group B sandy loam and Group C that had a mixture of clay and sandy clay
loam. Each area was instrumented with 12 soil moisture sensors (5TEs, DECAGON Devices, Inc., Pullman, US)
at 0.1 m soil depth, following a spatially optimized nested sampling scheme, as described by Lark (2011).
To test maximum Ɛ values in the field, areas near each site were wetted up at 0.1 m depth until ponding occurred
and Ɛ was measured in the saturated or ponded water using 2 5TE sensors. The sandier soils in site B had high



infiltration and therefore it was difficult to record ponded water at site B. It was also found that the soils at 0.1m
depth in site B even if it had been raining consistently, they remain unsaturated, therefore high in-situ VWC in site
B was considered to be near 0.4 m$^3$ m$^{-3}$. During dry conditions in the field, soil core samples were taken from 0.10
m depth and Ɛ was measured by 5TE sensors during the same time. The soil cores were weighed in the field to
measure soil moisture gravimetrically and then dried in the lab to measure the VWC at the time of *in-situ*
measurements.
### 3.2    5TE Sensor operation principles and acquisition
Like all capacitance probes, according to the 5TE manual (Decagon Devices, 2014) the 5TE sensor uses an
electromagnetic field to measure the dielectric permittivity (Ɛ) of the surrounding soil. It supplies a 70 Mhz
oscillating wave to the sensor prongs, which then charges depending on the dielectric of the surrounding soil. The
resulting charge is proportional to the soil dielectric and soil volumetric water content and the output value from
the 5TE microprocessor is a value of Ɛ from the sensor. The 5TE sensor also measures temperature and electrical
permittivity. The sensor dimensions are 10 x 3.2 x 0.7 cm, active measurement length is 5.2cm. It has volumetric
water content (VWC) accuracy ±3% using the Topp equation in typical mineral soils that have electrical
conductivity <10dS/m. VWC accuracy improves from ±1 to 2% in any porous medium using a soil-specific
calibration.
During field measurements within each area A, B and C (locations shown in fig. 1), the sensors were measured
every 15 minutes for 10 months (January 2012 to October 2013) and were logged using an Adcon telemetry
system (Klosterneuburg, Austria) which was linked to individual sensors using the SDI-12 serial communications
standard. Measurement data was stored locally at each of the eight nodes before being transferred over a radio link
to a local coordinator node where it was relayed to a GSM link for storage and processing within a relational
database hosted on a server at the British Geologcial Survey offices in Keyworth, UK.
During calibration an Em50 logger was used to acquire data from three 5TE probes via a Stereo to USB port, using
the ECH20 Utility software (Decagon Devices, 2016).
### 3.3    Soil shrinkage
Soil shrinkage of clay soils was measured using a technique developed by the British Geological Survey (BGS),
which uses a laser rangefinder to measure the height and diameter of a soil core, as it dries on a motorised rotating
platform (Hobbs et al., 2014). The laser scans up to 3600 points around the soil core periphery and then weighs
the soil after each scan. The volume reduction of the soil core is graphed against the core soil water content and
the shrinkage limit is taken at the last measurement, as described by Head (1992) and particle density was set at
2.65 mg/m$^3$. At the start of each measurement the clay cores were approximately at field capacity. Replicate cores
taken from Groups A and C were measured for shrinkage, but the sandy clay loam (Group B) was not measured,
because it loses structure once it begins to dry. Soil shrinkage was also measured for all soil cores that were oven
dried to provide percentage shrinkage from saturated to oven dried cores.





### 3.4 Water release curves

Three replicate soil cores taken from Groups A to C were saturated with deionised de-aired water for 2 to 10 days
and then equilibrated in a pressure plate chamber at metric potentials 0, -33, -50, -100, -250, -500, -1000 and
finally at -1500 kPa. Soil core mass was measured after each equilibration and used to calculate volumetric water
content. After each applied suction, shrinkage of core volumes was estimated by measuring the core volumes.
The model proposed by Van Genuchten (1980) (VG) was used as a basis for determining the shape of the water
release curves. The VG curve (Eq. (1) describes how VWC, $\theta$ changes as a function of the suction pressure,
$\psi$: $\theta(\psi) = \theta_r + \frac{\theta_s - \theta_s}{[1+(\alpha|\psi|)^n]^{1-1/n}}$       (1)
Where $\theta_r$ and $\theta_s$ are the residual and saturated VWC respectively and $\alpha$ and $n$ are dimensionless empirical
parameters. These four parameters were defined in two ways: Firstly using the information provided from the soil
texture information (fig. 2) to determine the parameters from the widely used ROSETTA pedotransfer function
software (Schaap et al., 2001). Secondly, we optimized the parameters to achieve the best fit to the observed data
using the unconstrained non-linear minimization procedure in MATLAB (fminsearch function).

### 3.5 Calibration of *in-situ* sensors in contrasting soil

We tested two methods to calibrate the Decagon 5TEs: a disturbed method and undisturbed method, which
measured soil from the three soil type areas A, B and C (shown in fig. 1).

### 3.5.1 Disturbed calibration method

The disturbed method consisted of taking five litres of soil from the field at 0.1m soil depth, air drying it and then
sieving the soil through a 2mm sieve. The clay and sandy clay were ground using a grinder to break down the soil
aggregates to less than 2 mm. The air-dried sieved material was then put into a bucket of known volume and
compressed to bulk densities that ranged from 0.8 to 1.1 cm$^2$/cm$^3$, depending on soil type at 0.1 m depth. The soil
moisture was then measured separately by two 5TE sensors (fig. 3a) to provide two values: the dielectric constant
of the soil ($\epsilon$) and volumetric soil water content (VWC) estimated from the DECAGON Devices factory calibration
for mineral soil, which uses the Topp equation, Eq. 2 (Topp et al., 1980).
$\theta = -5.3 \times 10^{-2} + 2.92 \times 10^{-2}\epsilon - 5.5 \times 10^{-4}\epsilon^2 + 4.3 \times 10^{-6}\epsilon^3$       (2)
$\theta$ is volumetric water content, $\epsilon$ is dielectric constant
A small core of soil was taken from the bucket to measure gravimetrically the VWC of the air dried soil. The
bucket was then emptied into a larger container and a quantity of water was added to make up approximately 10%
VWC. This water was mixed evenly into the dry soil and the bucket was repacked to a similar bulk density and
the $\epsilon$ and VWC were again measured using a 5TE sensor. Another small core of known volume was again taken
to measure gravimetrically the VWC of the new soil water content. These steps were repeated four times, until the
water content approached saturation. This method provided five data points of: gravimetrically measured VWC, $\epsilon$
and VWC measured by the 5TE probes and is a typical method for calibrating soil moisture sensors by Decagon



Devices, Inc. (Cobos and Chambers, 2010). A final VWC close to 100 % was taken by creating a small depression
in the middle of the bucket in saturated soil. Water was poured into the depression to create ponded water and the
5TE probe was placed in the ponded water to take readings of the Ɛ and VWC. A sample of the ponded water was
then taken to measure gravimetrically the percent of sediments in the ponded water. It was observed that once
VWC was over 50%, aggregate structures disappeared and its internal structure disappeared to form a soup-like
structure. This change in structure is shown in fig.3A.
**3.5.2     Undisturbed calibration method.**
The undisturbed method uses a soil core of diameter 19.5 cm and height 23 cm, where a 5TE sensor is placed at
0.10 m below the ground surface and the soil core is cut out of the ground and placed directly into a PVC core (fig.
3b), ensuring that the sensor remains undisturbed. The 19.5 ×23 cm core size ensures there are no edge effects
since the sensor is further than 5 cm from the core sides. The core was placed on a sand table and water was added
to allow the core to saturate. After five days of saturation, the core was then weighed and the Ɛ and VWC were
measured from the sensor placed in the core. The core was then allowed to air dry and every 5 to 10 days the core
was weighed, the volume estimated and the Ɛ and VWC were measured from the sensor in the core. After two
months of air drying, the core was put into a 30 °C oven to dry for a month. During this time measurements of core
weight and volume were taken and the sensor in the core continued to monitor Ɛ and sensor VWC. In this way a
drying curve of gravimetric VWC and sensor Ɛ and VWC were measured to provide a calibration curve. A final
measurement was taken in ponded water, by removing the 5TE probe and creating a depression in the centre of
the soil core. The depression was then filled with water and Ɛ and VWC were measured by inserting the 5TE probe
in the ponded water. The ponded water was also sampled to estimate the percentage sediments in the water.
**4     Results**
**4.1     Soil characteristics**
The electrical conductivity of saturated soil within the field had conductivities at 0.1 m depth from 0.001 to 0.12
dS/m in clay soils, 0.001 to 0.01 dS/m in sandy loam clay and 0.001 to 0.7 mdS/m in sandy clay soils. These
values are below the conductivity threshold of <10dS/m, which 5TE probes are considered to accurately measure
VMC (Decagon Devices, 2014).     The particle size distributions and associated soil texture classes for sites A to
C are shown in fig. 2. Site positions 1 to 4 (on the upper part of the slope, fig. 1) have very large clay contents
(ranging from 85 to 100%) and positions 5 to 7 (at the bottom of the slope, fig. 1) have relatively large sand
contents (between 20% to 67%), while position 8 (lower east side of the site slope) has the largest average amount
of silt (26%), a relatively small amount of sand (51 to 10%) and a larger clay content (up to 64%). The decreasing
clay content from the upper to lower slope, coincides with the Whitby Mudstone Formation and the lower Staithes
Sandstone Formation (Gunn et al., 2013). Bulk density ranged from 0.93 to 1.22 g cm$^{-3}$ in areas 1 to 4, 1.22 to 1.34
g cm$^{-3}$ in areas 5 and 6 and 1.36 to 1.51 g cm$^{-3}$ in areas 7 and 8.



### 4.2 Soil Shrinkage

Using the BGS method to estimate soil shrinkage limit for Group A was 13% and 16% and for Group C it was 9% and 14%. Shrinkage for these cores was drying from field capacity to oven drying at 30°C. Measuring the change of clay volumes from saturated soil water content to oven drying gave much higher shrinkage limit ranges such as 35% for Group A, and 4% for Group B and 23% for Group B. The shrinkage of clay soils for group A are comparable to some Dutch clay soils, which have been measured to have 42% shrinkage from saturation to a pressure head of -16000cm (Bronswijk and Eversvermeer, 1990).

During dry field conditions, the clay soils were observed to crack and create fissures up to 0.10 m in width and during prolonged wet conditions the soil was observed to expand and these fissures closed. During wet winter intervals rainfall was observed to pond within these fissures on the upper hillslope in the clay size-fraction dominated soils.

### 4.3 Water release curves and PTF

The maximum, minimum and average values for saturated VWC, for permanent wilting point (taken at -1500 kPa), field capacity (-33 kPa), and saturated VWC using the pressure chamber technique are given in Table 1.

The water release curves for the three different soil groups are shown in fig. 4. Here the VG curves using the ROSETTA parameters (green lines) underestimate the mean observed VWC (black dots) significantly, with biases between -0.12 and -0.22 cm$^3$/cm$^3$ (Table 2). The fitted VG curves (red lines in fig. 4) are much more reflective of the soil moisture characteristics of all of the soil types and show considerably smaller biases of 0.001 cm$^3$/cm$^3$ and root mean quadratic errors between 0.01 and 0.02 cm$^3$/cm$^3$. A comparison of the ROSETTA and fitted parameters in Table 1 shows that the ROSETTA pedotransfer function underestimates both the $\theta_r$ and $\theta_s$ parameters for all soil types resulting in large negative biases.

### 4.4 Dielectric constant, factory calibration and *in-situ* soil conditions

The data measured by the 12 5TE sensors in each area A, B and C were pooled together for each site and maximum, minimum and average VWC was estimated using the generic Topp equation and are plotted in fig. 5. The clay (site A) and sandy clay (site C), had the largest maximum and minimum Ɛ in comparison to the sand clay loam (site B), which had the smallest maximum and minimum Ɛ.

To test the range of VWC calculated by the Topp equation, we compared *in-situ* observations and measured VWC for each site using gravimetric methods. During dry field conditions, the clay and sandy clay sites were observed to form cracks up to 0.15 m wide and 0.20m deep. After prolonged heavy rainfall, some cracks were observed to be filled with water, which coincided with maximum *in-situ* Ɛ values for sensors in areas of clay cracking. Similar maximum Ɛ values were reached when 5TE sensors were in a slurry of ponded soil water both in disturbed and undisturbed soil cores (Table 3). Sandy clay soils (site B) however, did not crack and were observed to freely drain, causing no ponding water; this created lower *in-situ* maximum Ɛ values in site B, as shown in Table 3.



Using the Topp equation, the Ɛ values were converted to VWC for the three soil types, A, B and C and the
maximum, minimum and average VWC are shown in fig. 5. The saturated VWC (at 0 kPa), field capacity (at -33
kPa) and permanent wilting points (at -1500 kPa) measured by the water release curves, including known points
of *in-situ* VWC given in Table 3 were superimposed on fig. 5. Aligning the known *in-situ* VWC and the saturated,
field capacity and permanent wilting points, shows that the Topp equation under-estimates the range of measured
Ɛ for the three clay soil types.
**4.5    Calibration of the soil moisture sensors**
Calibration curves using cubic, square and linear regression statistics are shown in fig. 6, where the gravimetric
volumetric water content is shown on the y-axis and the Ɛ is measured by the 5TE sensors are on the x-axis. The
results of the regression statistics are given in Table 5, where the cubic model is (Eq. 3):
$b_0 + b_1 Ɛ + b_2 Ɛ^2 + b_3 Ɛ^3,$ (3)
the quadratic model is (Eq. 4):
$b_0 + b_1 Ɛ + b_2 Ɛ^2,$ (4)
And the linear model is (Eq. 5):
$b_0 + b_1 Ɛ,$ (5)
where, Ɛ is the dielectric constant, measured by the 5TE sensors at 0.1 m soil depth.
According to the adjusted $R^2$ (Miles, 2005), the cubic model was the best fit for all soil types and calibration
methods.
Fig. 7 shows the Ɛ measured by the 5TE sensors converted to VWC using the best-fit models for each disturbed,
undisturbed calibration methods and the Topp equation (eq. 1) for each soil type. The saturated VWC (at 0 kPa),
field capacity (at -33 kPa) and permanent wilting points (at -1500 kPa), and known *in-situ* VWC (given in Table
3) were superimposed on fig. 5 and 7
The largest differences between the calibration models occur for wet clay soils, where the disturbed calibration
method estimates maximum VWC values to be 0.31, 1.73 and 0.22 larger than the VWC values estimated by the
Topp equation using the same Ɛ *in-situ* values (shown in fig. 7) for clay, sandy clay loam and sandy clay
respectively. Whereas, the undisturbed method estimates maximum VWC values to be 0.38, 0.10 and 0.43 larger
than the VWC values estimated using Topp equation. The known *in-situ* VWC values for all soils (shown as
circles) coincide with the best-fit calibrations, showing that the undisturbed method of calibration increases the
VWC into the expected range of saturated and permanent VWC.
**5    Discussion**
The under-estimation of VWC of the clay-rich soils (groups A and C) at different matric potentials estimated by
the ROSETTA PTF in comparison to the water release curves developed from the pressure chamber method (fig.

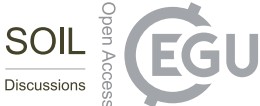

4) is likely due to the volumetric expansion of the clay-rich soil, under wet conditions, which is not taken into
account by the ROSETTA PTF. The high saturated VWC of the clay soil in particular for site A is possible because
of the relatively low bulk density and swelling of these soils under wet conditions, while soil pores remain water
filled at high matric potentials (-1500kPa), causing the fitted curves in fig. 4 to be positioned at larger water
contents than ROSETTA predicted PFTs. Such importance of characteristics of clay-rich soils to develop PTFs
are discussed by Hodnett and Tomasella (2002) and Gaiser et al. (2000). We therefore suggest that it is important
to use lab techniques to estimate water release curves of soil cores taken from the different soil types to develop
site specific PTFs for these clay-rich soils. This study corroborates the results from a study by Patil, et al. (2010)
who evaluated ROSETTA and concluded that it had limited capability to determine water retention functions for
shrink-swell soils in India and recommended region-specific PFTs to predict available water capacity.
The combination of the water release curves (Table 1 and fig. 4) and the observed *in-situ* VWC for wet and dry
intervals of three sites (Table 3) are important indicators to verify the generic factory calibration (in this case the
Topp equation) for the 5TE sensors. Investigation of Figure 5, shows that the generic factory calibration,
consistently underestimates the maximum, mean and minimum ranges of VWC in particular for sites A and C,
because the wettest VWC points using the generic factory calibration remained at field capacity, when the these
recorded wet VWC points were observed to be saturated in the field and the driest VWC are too dry, when *in-situ*
observations estimate soils to be near permanent wilting point. The sandy clay loam however, was not observed
to reach saturated VWC because these soils drained quickly and therefore the generic factory calibration was
considered sufficient to estimate VWC for site B.
Figure 6 shows that the undisturbed calibration provides the best calibration for all soils, because it is consistent
with observed *in-situ* field data and soil water release curves of soil cores measured from the three soil types. This
calibration is able to represent the large range of VWC occurring through wet and dry intervals.
The Topp equation and the disturbed calibration estimated smaller ranges of VWC, but the disturbed calibration
over-estimated VWC, while the Topp equation under-estimated VWC (figs. 7, A and C). The change in soil
structure when adding approximately 50% water to the ground and sieved sample during the disturbed calibration
method, is likely to cause the over-estimation of VWC. The grinding and sieving to < 2mm of the clay-rich samples
removed  the structure from the soil cores, creating large void ratios and a weak bonded structure, which is typical
of "quick" clays, which have a propensity to liquefy and flow (Gauthier and Hutchinson, 2012). This change in
soil structure is likely to cause lower $\varepsilon$ values at VWC >50% (fig. 6), creating calibration curves that over-estimate
VWC in comparison to the soil structure of the Undisturbed calibration method as shown in fig. 3.
**6   Conclusions**
The volumetric expansion of clay-rich soils under wet conditions is an important characteristic that is not always
taken into account using PTF, which can cause under-estimation of VWC during wet conditions. This study shows
that grinding and sieving clay soils to <2mm and then repacking the clay to bulk densities similar to *in-situ* field
bulk densities does not represent the same field conditions for accurate calibrations to convert $\varepsilon$ to VWC. When
adding >50% water to the ground and sieved soil samples, $\varepsilon$ values to VWC >50% were observed to be lower than



using undisturbed soil cores taken from the field. Further studies are needed to understand the causes of the loss
of structure in this soil and its relationship with Ɛ.
Generic factory calibrations for most soil sensors have a range of measurement from 0 to 50%, which is not
appropriate for the studied clay-rich soil, where ponding can occur during persistent rain events, which are
common in temperate regions. The range of VWC from saturated (0 kPa) to permanent wilting point (-1500 kPa)
were 36 to 72%, 12 to 50% and 31 to 69% for clay, sandy clay loam and sandy clay respectively. *In-situ*
observations did not always reach saturation, such as in the sandy clay loam site, but saturation was reached in the
clay and sandy clay sites. Other studies have also concluded that the precision of capacitance sensors, worsen in
saturated soils (Evett et al 2006). Therefore it is important to know the range of Ɛ values that the sensor is measuring
in field conditions to ensure that the conversion to VWC effectively provides actual VWC, rather than simply
taking soil from the field and repacking it to similar field bulk density in lab conditions.
**Acknowledgements**
The authors thank Mr and Mrs Gibson for the access to their land. This paper is published with the permission of
the Executive Director of the British Geological Survey (Natural Environment Research Council) and was funded
by the British Geological Survey, NERC. We are grateful for Matt Kirkham's help and advice during laboratory
analysis.



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





**List of tables**
Table 1. Minimum, average and maximum saturated (0 kPa) water content, field capacity (-33 kPa) and wilting
point (-1500 kPa), estimated from pressure chamber data, for s A, B and C. Standard deviation of averages for
three replicates are given in brackets.
Table 2. Soil release curve parameters for the VG model estimated by fitting (bold) and using the ROSETTA
pedotransfer function (square brackets).
Table 3. Maximum (Max.) and minimum (Min.) dielectric constant ($\mathcal{E}$) measured by the 5TE sensors *in-situ* for
three groups, A, B and C. Gravimetric volumetric water content (VWC) is the soil water fraction measured
gravimetrically in relation to maximum and minimum $\mathcal{E}$.
Table 4. Maximum (Max.) and minimum (Min.) dielectric constant ($\mathcal{E}$) measured by the 5TE for the disturbed and
undisturbed calibration methods for three soil types, A, B and C. Gravimetric volumetric water content (VWC) is
the soil water fraction measured gravimetrically in relation to maximum and minimum $\mathcal{E}$.
Table 5. Linear (L), quadratic (Q) and cubic (C) models for predicting SWC using the dielectric constant measured
by the 5TE sensors. "Undist" is the Undisturbed calibration method and "Dist" is the Disturbed calibration method.
The $R^2$ and Adjusted (Adj.) $R^2$ for each model are included and the greyed lines indicate the model with the highest
Adj $R^2$. These values were used to estimate VWC in fig. 7.
**List of figures**
Fig. 1. Dominant soil texture classes of the study area. Red dots indicate positions of soil moisture sensors. Hatched
yellow lines are recent flow deposits and blue hatched lines are relict flow deposits (Merritt et al., 2014)
Fig. 2.UK Soil Survey of England and Wales texture triangle, which considers a silt − sand limit of 60 µm.
Abbreviations: Cl is Clay, SaCl is sandy clay, ClLo is Clay loam, SiClLo is Silty clay loam, SaClLO is Sand clay
loam, SaLo is Sandy Loam, SaSiLo is Sandy silt loam, SiLo is Silt loam, LoSa is Loamy sand, Sa is Sand. The
letters show the soil texture for three soil texture groups relating to fig. 1.
Fig.3. A) An aerial view of the approach in which soil structure is removed. Upper image is VWC near 30% and
the lower image is VWC > 55%. B) The undisturbed method. The soil core taken is out of the PVC core.
Fig. 4) Soil moisture release curves predicted using the VG model with parameters from the ROSETTA
pedotransfer function (green curves) and parameters optimised to the observed data (red lines). Mean observed
VWC (black dots) were calculated from three replicate cores (grey dots).
Fig. 5) Maximum, minimum and mean VWC calculated by generic factory calibration Topp equation) and hourly
rainfall intensity (grey bars). Maximum and minimum saturated VWC, average field capacity and permanent
wilting point values estimated from water release curves are plotted. The blue circles indicate maximum in-situ
VWC when soil was saturated and the red circles indicate known minimum in-situ VWC A is clay soil, B is sandy
loamy clay soil and C is sandy clay.



1 Fig. 6) Estimated linear, quadratic and cubic models showing the relationship against VWC and dielectric constant

2 for (a) Group A (clay), (b) Group B (sandy clay loam) and (c) Group C (sandy clay).

4 Fig. 7) Estimated VWC for *in-situ* Ɛ data measured during 2012 for three different soil types (a) Group A (clay),

5 (b) Group B (sandy clay loam) and (c) Group C (sandy clay). The three calibration models are from disturbed

6 and undisturbed methods and the factory calibration is using the Topp equation.



1    Table 1.

| Hydrological characteristic | | A | B | C |
|---|---|---|---|---|
| Saturated VWC | Minimum | 68 | 44 | 64 |
| Saturated VWC | Average | 72 (±4.41) | 50 (±9.99) | 68 (±4.44) |
| Saturated VWC | Maximum | 77 | 61 | 72 |
| Field Capacity VWC | Minimum | 53 | 27 | 51 |
| Field Capacity VWC | Average | 54 (±0.48) | 30 (±4.07) | 52 (±2.32) |
| Field Capacity VWC | Maximum | 54 | 35 | 55 |
| Wilting point VWC | Minimum | 36 | 12 | 31 |
| Wilting point VWC | Average | 38 (±2.37) | 14 (±2.29) | 34 (±2.93) |
| Wilting point VWC | Maximum | 40 | 17 | 36 |



2
3    Table 2.

| Soil Group | $\theta_r$ (cm³/cm³) | $\theta_s$ (cm³/cm³) | $\alpha$ (-) | $n$ (-) | RMSE (cm³/cm³) | Bias (cm³/cm³) |
|---|---|---|---|---|---|---|
| A | **0.33** [0.10] | **0.72** [0.46] | **0.026** [0.015] | **1.28** [1.25] | **0.01** [0.22] | **0.002** [-0.22] |
| B | **0.11** [0.04] | **0.50** [0.39] | **0.027** [0.027] | **1.32** [1.45] | **0.02** [0.12] | **0.002** [-0.12] |
| C | **0.21** [0.06 – 0.10] | **0.69** [0.39 – 0.46] | **0.030** [0.015 - 0.021] | **1.18** [1.25 – 1.42] | **0.02** [0.19 – 0.28] | **0.002** [-0.28 - -0.19] |



1    Table 3

| Soil | Min. Ɛ | Gravimetric VWC (min) | Max. Ɛ | Gravimetric VWC (max.) |
|------|--------|-----------------------|--------|------------------------|
| A | 9.8 | 0.33 ±0.04 | 53.4 | Ponded water |
| B | 5.0 | 0.15 ±0.02 | 31.7 | 0.37 ±0.03 |
| C | 5.6 | 0.29 ±0.05 | 45.0 | Ponded water |

3    Table 4

| Soil | Calibration Method | Min. Ɛ | Gravimetric VWC (min) | Max. Ɛ | Gravimetric VWC (max.) |
|---|---|---|---|---|---|
| A | Undisturbed | 4.12 | 0.06 | 51.0 | 1 |
| A | Disturbed | 3.34 | 0.06 | 46.3 | 0.96 |
| B | Undisturbed | 5.02 | 0.08 | 56 | 1 |
| B | Disturbed | 2.72 | 0.01 | 48.6 | 0.97 |
| C | Undisturbed | 4.88 | 0.08 | 44.8 | 1 |
| C | Disturbed | 3.48 | 0.06 | 40.0 | 0.95 |



1     Table 5.

| Method | Soil | Model. | $b_0$ | $b_1$ | $b_2$ | $b_3$ | $R^2$ | Adj $R^2$ |
|--------|------|--------|-------|-------|-------|-------|-------|-----------|
| Undist | A | L | 0.04215 | 0.01774 | NaN | NaN | 0.9290 | 0.9200 |
| UnDist | A | Q | -0.01233 | 0.02265 | -0.00009 | NaN | 0.9340 | 0.9208 |
| Undist | A | C | -0.19461 | 0.05199 | -0.00135 | 0.00002 | 0.9632 | 0.95274 |
| Dist | A | L | 0.10222 | 0.02212 | NaN | NaN | 0.8379 | 0.8129 |
| Dist | A | Q | -0.04726 | 0.04583 | -0.00053 | NaN | 0.9361 | 0.9201 |
| Dist | A | C | -0.21530 | 0.09012 | -0.00308 | 0.00004 | 0.9645 | 0.95162 |
| Undist | B | L | 0.00528 | 0.01848 | NaN | NaN | 0.9737 | 0.9701 |
| Undist | B | Q | -0.06405 | 0.02524 | -0.00012 | NaN | 0.9852 | 0.9820 |
| Undist | B | C | -0.19958 | 0.04780 | -0.00111 | 0.00001 | 0.9943 | 0.99252 |
| Dist | B | L | -0.00510 | 0.02562 | NaN | NaN | 0.9721 | 0.9675 |
| Dist | B | Q | -0.05841 | 0.03515 | -0.00026 | NaN | 0.9870 | 0.9835 |
| Dist | B | C | -0.10161 | 0.04869 | -0.00111 | 0.00001 | 0.9898 | 0.9857 |
| Undist | C | L | 0.04032 | 0.02163 | NaN | NaN | 0.9457 | 0.9366 |
| Undist | C | Q | -0.07330 | 0.03339 | -0.00024 | NaN | 0.9664 | 0.9571 |
| Undist | C | C | -0.26734 | 0.06950 | -0.00208 | 0.00003 | 0.9851 | 0.9792 |
| Dist | C | L | 0.09374 | 0.02209 | NaN | NaN | 0.8760 | 0.85696 |
| Dist | C | Q | -0.05660 | 0.04489 | -0.00049 | NaN | 0.9839 | 0.97986 |
| Dist | C | C | -0.14662 | 0.06890 | -0.00186 | 0.00002 | 0.9898 | 0.9861 |



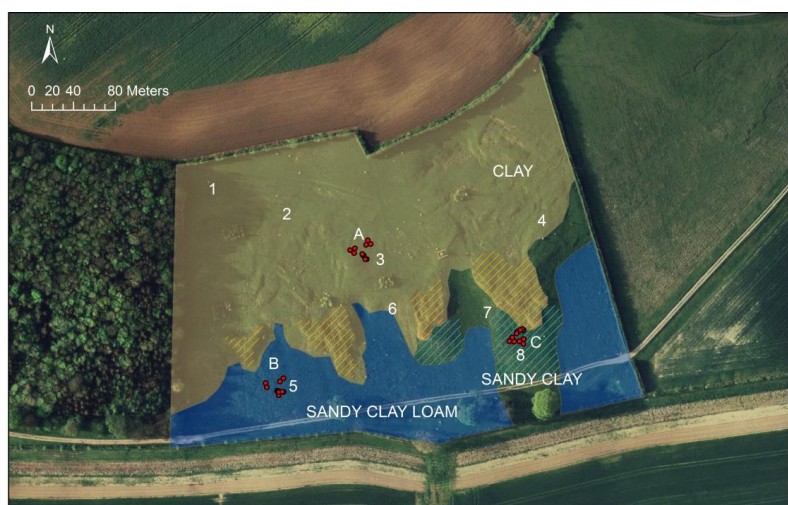

3    Figure 1



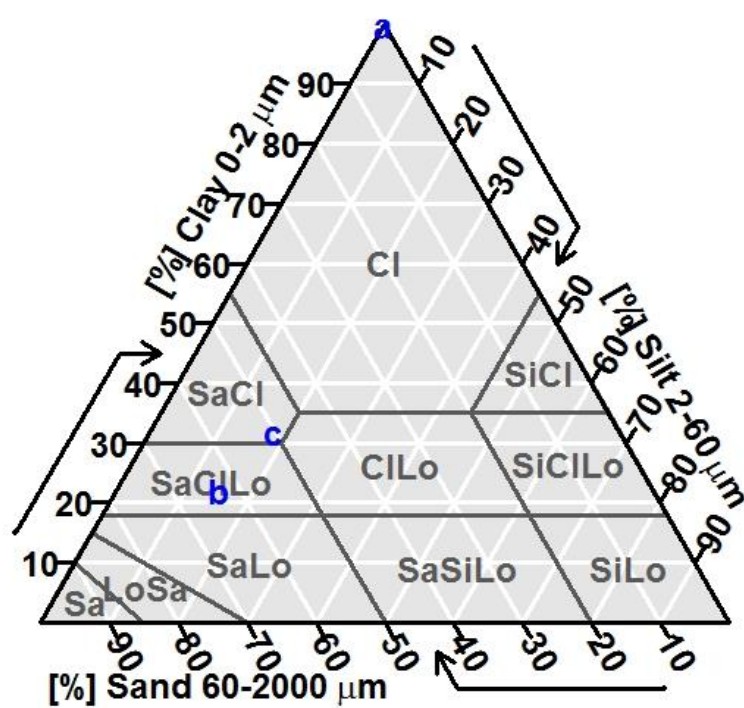

4      Figure 2




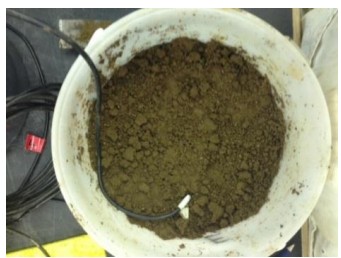

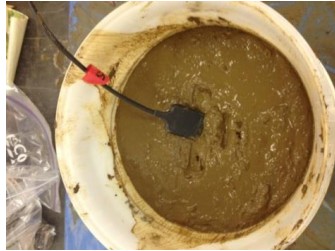

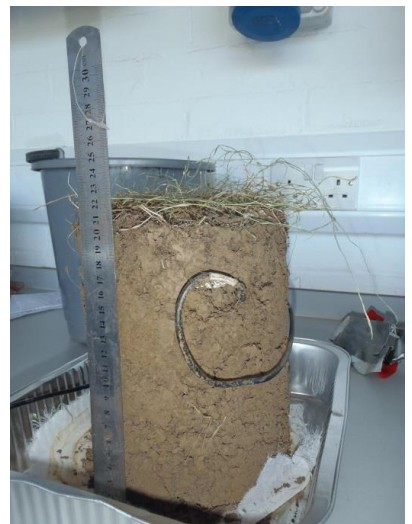

A                                         B

1       Figure 3)



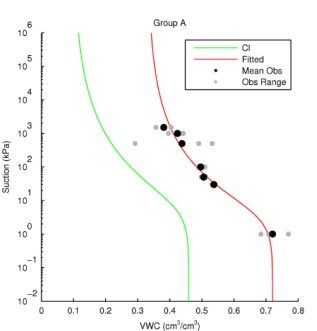 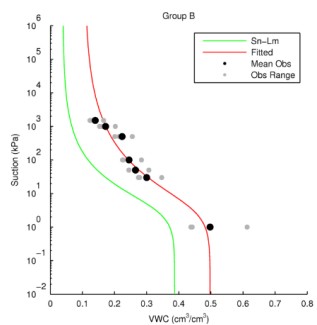 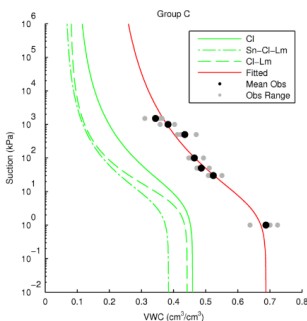

2      Figure 4)




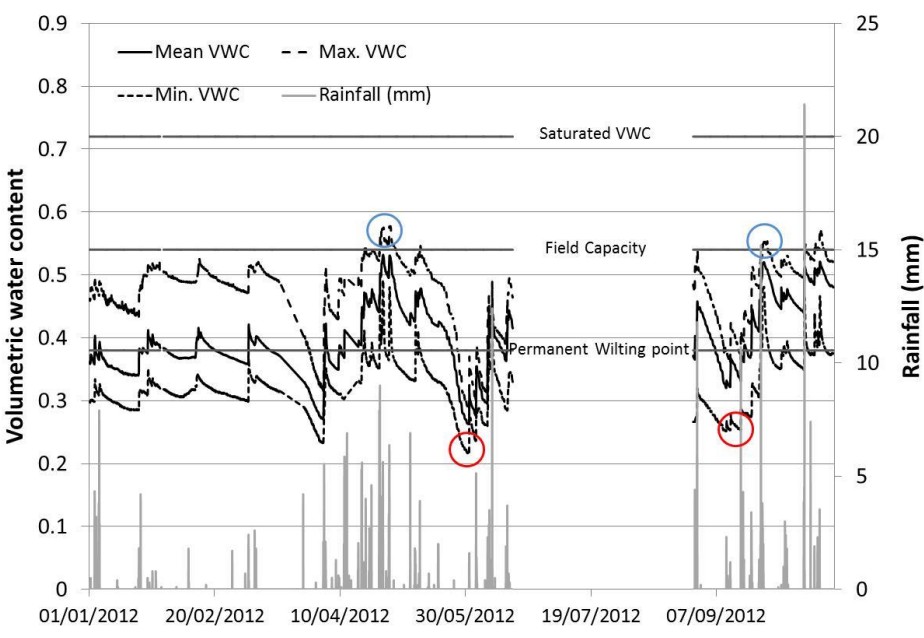

3          (a)

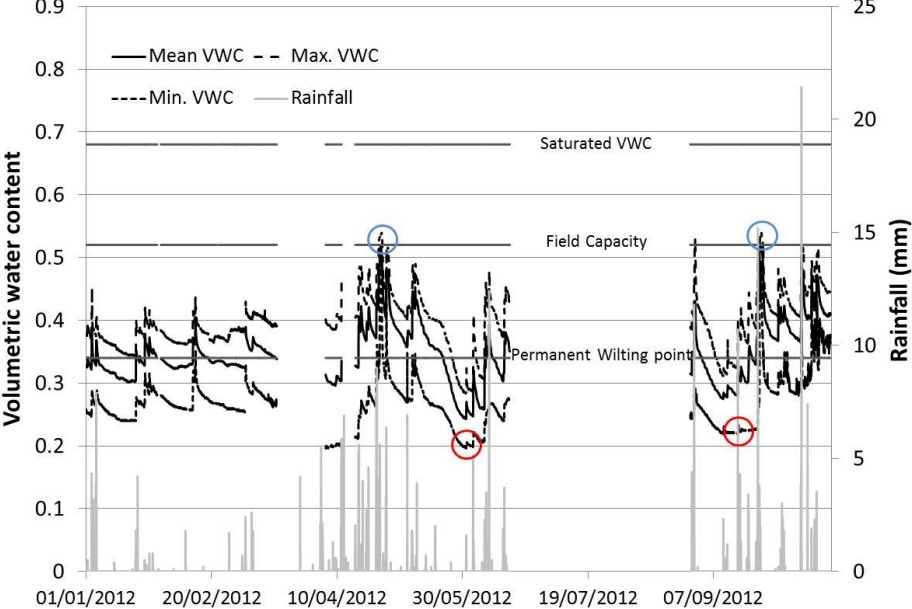

7          (b)



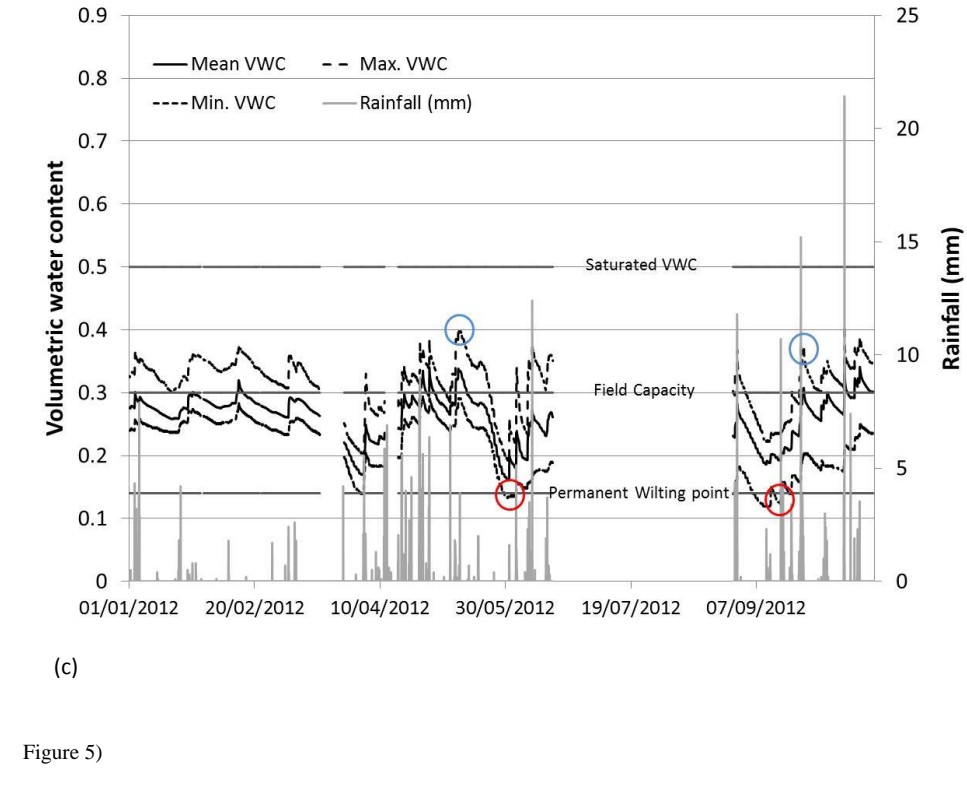

3    (c)

5    Figure 5)



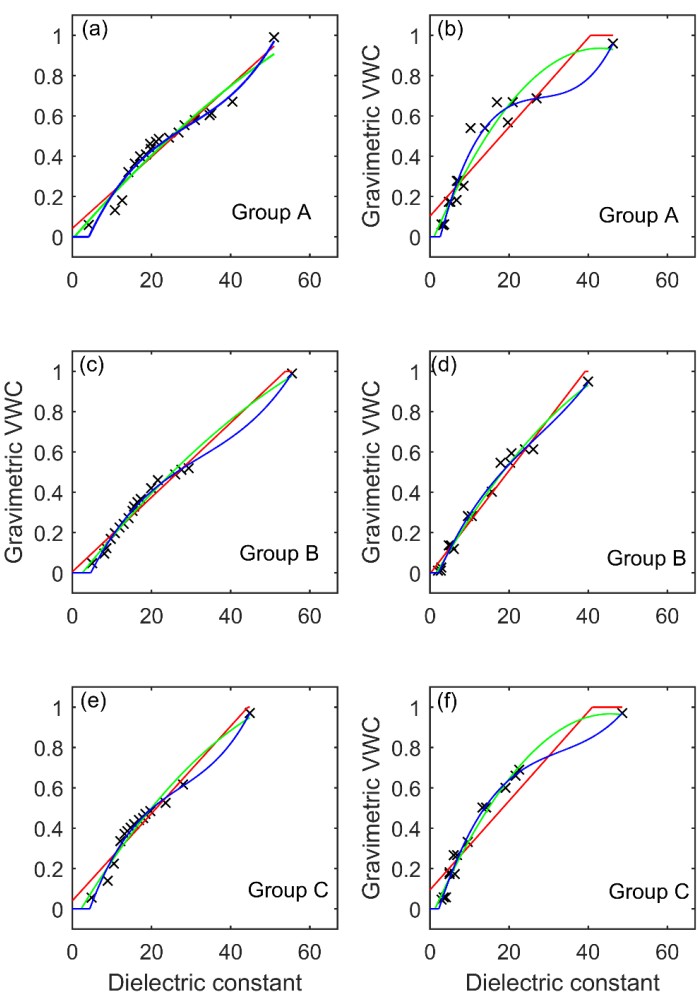

2    Figure 6)




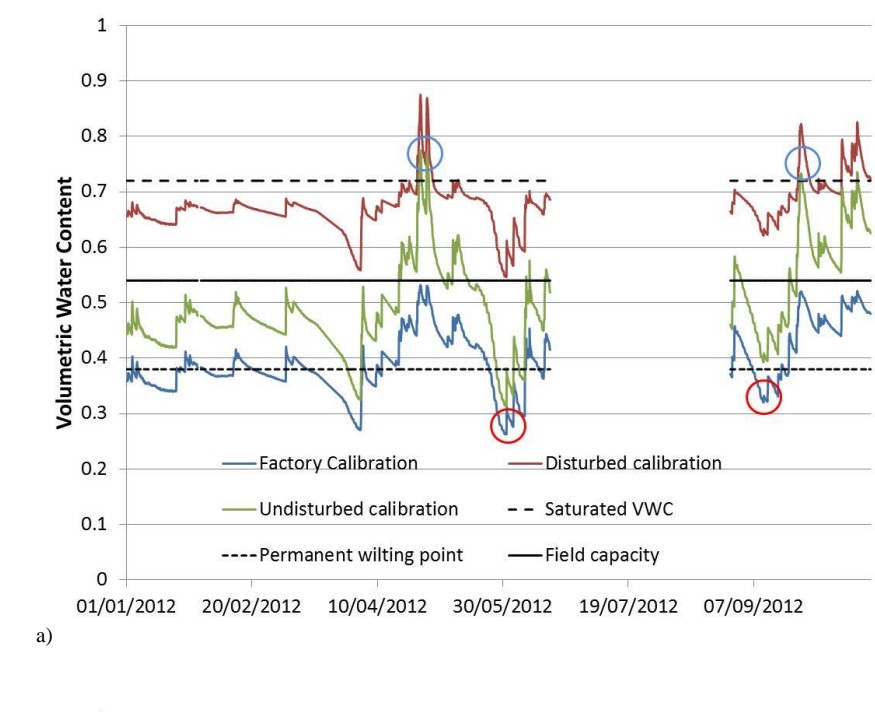

2   a)
3

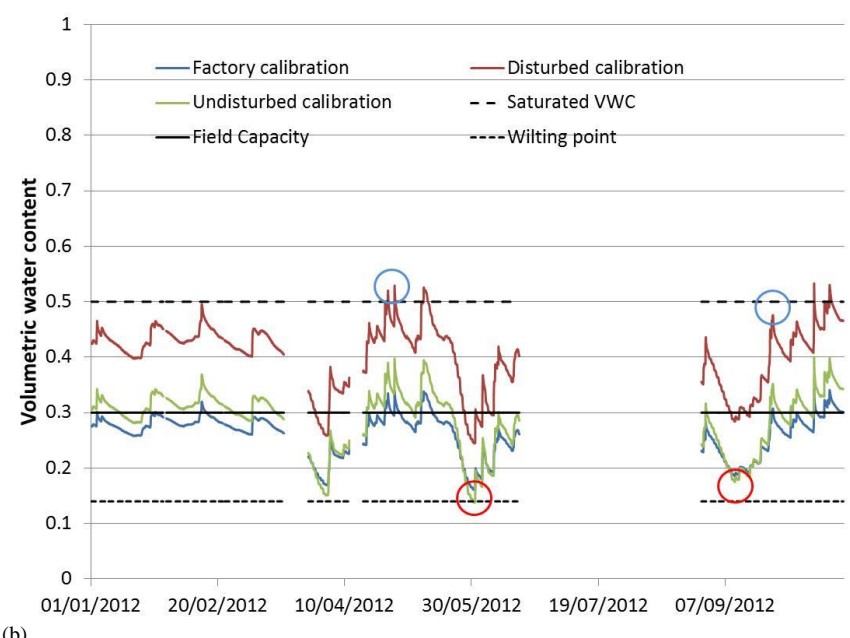

6   (b)





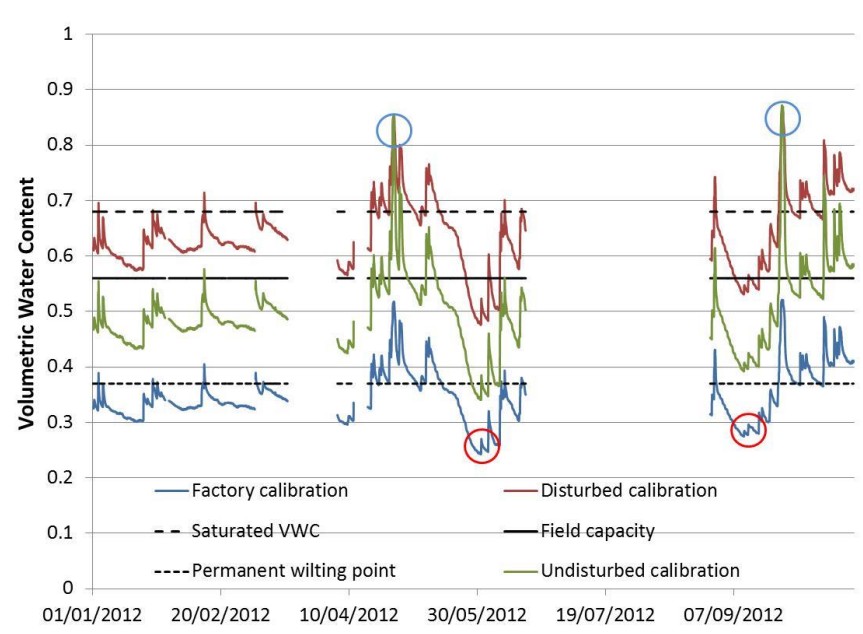

2    (c)
3
4    Figure 7)