# Peer review of "Approaches to calibrate *in-situ* capacitance soil moisture sensors and some of their implications"

_SOIL, 2016_

## Referee Comment (RC1) · Anonymous Referee #1 · 11 Jul 2016

The manuscript lacks focus and conciseness, while at the same time serious doubts arise about the data quality and the validity of the results, discussion and conclusions. For this reason I can only recommend rejection. A list of general and specific comments, by no means exhaustive, is provided in support of this recommendation and of the authors, in case they want to rework their manuscript.

General comments:

1. Lack of focus: What is the central topic of this manuscript? Calibration of soil moisture sensors? Too much emphasis is put on the description of the soils and the geology of the experimental site. Also the use of PTFs is clearly beyond the context of this manuscript. Too much non-relevant information is provided so that it becomes difficult for the reader to follow the rationale and figure out what the authors want to

achieve or what they have done. The same holds for the sections on soil shrinkage, since this information is later on not used (or it is not explained to the reader where it is used), or the use of a PTF to estimate water retention curves, since they have measured water retention data. A relevant central topic could be a comparison of calibration procedures with undisturbed and disturbed soil. The reader is expecting an analysis of both procedures, but only time series of measured soil moisture are provided.

2. A large body of literature is available on calibration and accuracy issues with soil moisture sensors, which should be taken into account. Several relevant studies are not cited. This information needs to be summarized and provided in this manuscript, instead of providing information from the user's manual. Clear knowledge gaps need to be identified from a thorough literature study before choosing the central topic of the manuscript . Since this manuscript deals with heavy clay soils, special emphasis has to be put on such soils. Their dispersive nature causes well known issues when measuring soil moisture with low-frequency devices. Therefore, the use of capacitance sensors to measure "permitivitiy" can be considered at least controversial.

3. When reading through the manuscript, serious doubts arise about the quality of the data and/or soundness of the used methods (e.g. comparing the weight of the soil column with water content measured at 0.1 m from the top of the column). A clay content of 100%, water contents near 1 or measured water retention values larger than the soil porosity are only some examples. Discussions and conclusions bases on these results are invalid.

Specific results:

P1 L32-L4: Capacitance probes provide an output signal (usually a voltage, RAW) which is then related to volumetric soil moisture. I think it is somewhat misleading to state that dielectric permittivity is determined using capacitance probes (See for example Vaz et al. (2013), and references therein)

P1 L34: The CS658 is the probe type that is used with the HydroSense system. This is not a capacitance probe. It measures travel time along a waveguide which is related to permittivity and soil moisture content.

P2, L2-9: Instead of listing this information from user manuals, it might be more informative to provide a summary of accuracies that were observed in other studies (e.g. Vaz et al. (2013); RoTimi Ojoand et al. (2015), and other work) for field an laboratory calibration

P2 L37- P3 L6: It is not clear what the use of PTFs can add to the evaluation of the field and/or laboratory accuracy of soil moisture sensors. I suggest to remove the sections on PTF′s from this manuscript.

P3 L18-20: Is this information relevant for the work at hand? I don′t think so. Id for L21-31. Summarize this information in a short statement.

P4, L10: How many replicates did you take at each of the eight points? If you did not take any replicates omit "replicate"

P4, L12: 100% clay? Correct.

P4, L10-17: The provided soil information should be limited to the positions of the sensor clusters (points 3, 5, and 8).Information on the other points and Fig. 2 is redundant and should be omitted.

P4, L18-31. This information is beyond the scope of the manuscript. Omit this paragraph.

P4, L36-P5, L6: Measuring permittivity with capacitance sensors is not a good practice. Omit references to permittivity throughout the manuscript. The 5TE sensors provide an output voltage which is then related to volumetric soil moisture.

P5, L4: Provide dimensions/volume of soil cores

P5, L8-16: Do not put emphasis on measurement of permittivity and summarize information provided by manufacturer. Instead, it would be more informative for the readers to compare with what others have found when performing independent tests with these sensors. Were the sensors installed horizontally or vertically? Where they completely buried? This is important information since measurements and accuracy will depend on this.

P5, L25-34: Is this relevant for the remainder of the manuscript. Where is this information used? Omit or indicate clearly where and how this information is used.

P6, L22: Fig 3A shows only one 5TE sensor. How were the sensors inserted? Vertically?

P7, L1: VWC of 100%? Correct: "VWC near saturation"

P7, L3-4: Is this kind of information required? I think it is redundant, clearly beyond the topic of the manuscript. This kind of details only distract the reader and do not contribute to a better understanding of the main topic of the manuscript. Omit this sentence.

P7, L8-20: Explain better. The sensor was inserted at a depth of 0.10 m? To be in accordance with the analysis for disturbed soil (0-0.10 m), should the sensor be inserted at 0.05 m depth? Was the sensor installed with the flat side of the prongs horizontally or vertically?

P7, L8-20: The main problem with this procedure is that the soil moisture at 0.10 m is not necessarily representative of the entire core. Sensors should have been installed at different depths to check this. Also is it improbable that saturation of a 23-cm high soil sample can reached by bottom-up moistening with a sand table.

P7, L17-20: This procedure is methodologically flawed and cannot easily be reproduced. Omit this measurement. Also, where in the remainder of the manuscript is this information used?

P7, L23-33: Provide this information in table format, but only for the locations where

the calibrations were performed. Just provide information for A, B and C. There is no need to provide a full description of the spatial distribution of soil properties or the geomorphology of the site. 100% clay seems quite suspicious to me. Could this be an artefact as a result of the used laboratory procedure?

P8, L13-21: Table 1 – Include units. Given the large difference in clay contents between soil A and C, larger differences in VWC are expected than those shown in Table 1. Overall, comparison with retention data provided in standard text books shows that the values reported here are way too high. Porosities calculated from the bulk densities provided on page 7 range from 0.65 to 0.53 for soil A, from 0.53 to 0.50 for soil B and from 0.50 to 0.43 for soil C. Water retention cannot be higher than this physical limit. This indicates systematic errors in the determination of the water retention and explains also the large differences with the PTF estimated water retention shown in Fig. 4.Which bulk density was used to calculate VWC from the gravimetric water content? In any case, use of PTFs is not recommended for extreme textures such as the one of soil A.

P8, L23-P9,L6: These results are invalid since the water retention data are wrong

P9, L9: "Gravimetric volumetric water content"? Water content can be expressed either gravimetrically or volumetrically. Water contents shown in Fig. 6 are wrong because neither gravimetric nor volumetric water content can reach such large values. The upper physical limit of water content is determined by the soil's porosity.

P9, L18. The cubic model contains 4 parameters (as compared to the 3 and 2 parameters of the other models), so it is rather obvious that the cubic model provided the best fit. To compare models with different numbers of parameters the Akaike Information Criterion or similar should be used.

The above comments lead to serious doubts about the validity of the results, discussion and conclusions provided in the remainder of the manuscript.

References:

Vaz, C.M., Jones, S., Meding, M. and Tuller, M., 2013. Evaluation of standard calibration functions for eight electromagnetic soil moisture sensors. Vadose Zone J., 12, doi:10.2136/vzj2012.0160.

RoTimi Ojo, E., Bullock, P. R., & Fitzmaurice, J., 2015. Field performance of five soil moisture instruments in heavy clay soils. Soil Sci. Soc. Am. J, 79:20-29.

---

## Referee Comment (RC2) · Anonymous Referee #2 · 17 Sep 2016

The manuscript entitled "Approaches to calibrate in-situ capacitance soil moisture sensors and some of their implications" by Archer et al provides a poor evaluation of some calibration approaches for soil moisture sensors. In the revision process some critical concerns have been detected:

- The innovation of the study is very limited; the calibration and validation of soil moisture sensors is a topic very well analyzed from long time ago. In fact, some publications related with this topic have been included by the authors in the introduction section but later, in the discussion section these were not considered. Equally, the consideration of disturbed and undisturbed soil for calibration has been analyzed previously in numerous studies. Thus, the major results obtained in this study are well-known for the scientific community. For these reasons the authors have to describe the main advances generated with this study and to carry out a full comparison with the previously

obtained results in other studies.

- The description of the methodology considered in the study is too long and descriptive. Thus, Section 3 provides a long explanation of general procedures that must be summarized referring to general publications. Even some sections included in Results could be moved to Material and Methods section (such as 4.1. Soil characteristics, or 4.5. Calibration of the soil moisture sensors).

- As opposite to the Material section, the results obtained in the study are very limited, repetitive and non-relevant to build a scientific publication in an international journal. Thus, the main results are concentrated in Section 4.4 and 4.5 and uniquely show volumetric water content curves and a simplified validation procedure.

- Discussion section is very limited. Numerous previous studies related with the topic of this study have not been considered. Here, the advances obtained compared with previous studies must be highlighted.

Additional comments:

Abstract: Some sentences are too long and difficult to understand. Please improve it.

Page 2 / Line 21: The reference Luis Gabriel et al., 2010 is not correct. It must be replaced by Gabriel et al., 2010

Page 2 / Lines 27-36: These paragraphs must be located at Discussion section

Page 4 / Line 5: Some spelling errors have been found ("calibration")

Page 4 / Line 15: In scientific publications "data" must be considered as plural and then sentences such as "This data was" must be modified.

Page 7 / Lines 23-30: Some descriptive parts must be moved to Material and Method section

Page 7 / Lines 30-33; Page 8 / Lines 5-7: The references to other studies must be

included in the Discussion section.

Page 9 / Lines 11-17: These sentences must be located at Material and Method section.

Page 9 / Lines 18-19: The procedure for estimating adjusted R2 must be explained in the Material and Method section

Page 11 / Lines 8-11: The references to other studies must be included in the Discussion section.

Page 13 / Line 1: The correct reference is "Gabriel JL, Lizaso . . ..". Please correct.

As conclusion, this manuscript in the current form has serious limitations and then, the recommendation provided by this reviewer is "Reject".

---

## Editor Comment (EC1) · J. A. Gomez (Editor) · 28 Sep 2016

My overall impression of this manuscript agrees with those presented by the two reviewers. The manuscript addresses an issue that have been covered by several studies previously and the authors have not been able to highlight in detail which are the new findings of their study on relation with the current knowledge in the field. Additionally there are several shortcomings in the manuscript that have been addressed by the reviewers in their interactive comments. For those reasons my recommendation is that this manuscript should be rejected for publication in soil.

I summarize below the major comments of the interactive discussion by the two reviewers in case the authors consider that they could rework their manuscript for a possible submission in the future.

[Figure]

1- Lack of focus: It seems that the topic of this manuscript is calibration of soil moisture sensors, but too much emphasis is put on the description of the soils and the geology of the experimental site and on the use of PTF (which probably contributes little to the manuscript).

2- It is not clearly explained what their manuscript contributes to previous studies on the same subject. The manuscript will benefit if the authors could highlight this clearly in the introduction and discussion section.

3- There is a large body of literature is available on calibration and accuracy issues with soil moisture sensors, but only part of this is taken into account. Additionally most of the previous that are studies are cited are in the introduction section, but the authors do not used them in the discussion of their results.

4- The overall structure of the manuscript is unbalanced. The introduction and material and methods sections are too long, while the results and discussion section are extremely concise.

5- There are several points in the manuscript where the authors might think about the soundness of the methods and the quality of some results. For instance: a) The soil moisture at 0.10 m is not necessarily representative of the entire core; b) the Akike (or any similar criteria) to evaluate the effect of fitting models with different number of parameters,